# Resonant exciton transfer in mixed-dimensional heterostructures for overcoming dimensional restrictions in optical processes

N. Fang [1] ✉, Y. R. Chang [1], D. Yamashita[2,3], S. Fujii [2,4], M. Maruyama [5], Y. Gao[5], C. F. Fong [1], K. Otsuka [1,6], K. Nagashio [7], S. Okada [5] & Y. K. Kato [1,2] ✉

Nanomaterials exhibit unique optical phenomena, in particular excitonic quantum processes occurring at room temperature. The low dimensionality, however, imposes strict requirements for conventional optical excitation, and an approach for bypassing such restrictions is desirable. Here we report on exciton transfer in carbon-nanotube/tungsten-diselenide heterostructures, where band alignment can be systematically varied. The mixed-dimensional heterostructures display a pronounced exciton reservoir effect where the longer-lifetime excitons within the two-dimensional semiconductor are funneled into carbon nanotubes through diffusion. This new excitation pathway presents several advantages, including larger absorption areas, broadband spectral response, and polarization-independent efficiency. When band alignment is resonant, we observe substantially more efficient excitation via tungsten diselenide compared to direct excitation of the nanotube. We further demonstrate simultaneous bright emission from an array of carbon nanotubes with varied chiralities and orientations. Our findings show the potential of mixed-dimensional heterostructures and band alignment engineering for energy harvesting and quantum applications through exciton manipulation.

The emergence of low-dimensional materials, encompassing one-dimensional (1D) carbon nanotubes (CNTs) and two-dimensional (2D) transition metal dichalcogenides (TMDs), has resulted in the discovery of unique physical phenomena not seen in bulk materials[1–4]. The reduced screening of Coulomb interactions in these materials leads to tightly bound excitons consisting of electron-hole pairs that remain stable at room temperature, playing a central role in optical processes[5–8]. In CNTs, the 1D electronic structure gives rise to van Hove singularities within the density of states, consequently forming discrete excitonic transitions. The resultant excitons are robustly confined within the 1D channel, exhibiting small Bohr radii[9], short excitonic radiative lifetimes[10], extended diffusion lengths[11], efficient exciton-exciton annihilation[12], and single-photon emission at room temperature[13].

While CNTs offer many intriguing luminescence properties due to their 1D nature, this dimensionality also imposes strict limitations on

[1]Nanoscale Quantum Photonics Laboratory, RIKEN Cluster for Pioneering Research, Saitama 351-0198, Japan. [2]Quantum Optoelectronics Research Team, RIKEN Center for Advanced Photonics, Saitama 351-0198, Japan. [3]Platform Photonics Research Center, National Institute of Advanced Industrial Science and Technology (AIST), Ibaraki 305-8568, Japan. [4]Department of Physics, Keio University, Yokohama 223-8522, Japan. [5]Department of Physics, University of Tsukuba, Ibaraki 305-8571, Japan. [6]Department of Mechanical Engineering, The University of Tokyo, Tokyo 113-8656, Japan. [7]Department of Materials Engineering, The University of Tokyo, Tokyo 113-8656, Japan. ✉e-mail: nan.fang@riken.jp; yuichiro.kato@riken.jp

the optical excitation process. Spatially, the diameter of a single-walled CNT is ~1 nm, which is considerably smaller than the diffraction-limited diameter of ~1 $\mu$m for incident laser light. This pronounced size difference restricts the interaction area between CNTs and light, inherently leading to a limited number of photons absorbed[14]. Spectrally, the isolated $E_{11}$ and $E_{22}$ transitions in the absorption spectrum necessitate wavelength-tunable lasers to excite CNTs of the desired chirality[15]. Furthermore, the 1D confinement of excitons in CNTs results in nearly perfect linear excitation polarization dependence, requiring the polarization angle to be aligned with the nanotube axis[16].

These limitations can be largely mitigated in higher-dimensional materials, such as 2D materials[17]. A much more efficient process, therefore, would involve connecting excitation in 2D materials with emission from 1D CNTs via exciton transfer, a phenomenon frequently observed in molecular complexes, biological systems, and semiconductor heterostructures[18–21]. High-quality interface needed for efficient coupling can be constructed by taking advantage of the van der Waals nature, as the 1D and 2D materials can be stacked together to form mixed-dimensional heterostructures without any lattice matching constraints[22,23]. With the ideal van der Waals interface, the band alignment in the heterostructure should be mainly governed by the band structure of each individual material rather than the interface, as is the case in the Anderson model[24,25]. The chirality-dependent bandgap of CNTs thus introduces an important degree of freedom into this system to engineer the band alignment, potentially enabling effective modulation of the exciton transfer process.

Here we investigate the exciton transfer process in CNT/tungsten diselenide (WSe$_2$) mixed-dimensional heterostructures. Compared with conventional $E_{22}$ excitation in CNTs, the WSe$_2$-based excitation offers an immensely larger absorption area, broader spectral response, and polarization-independent efficiency, all of which stem from the 2D nature of the material. Following an extensive exploration of CNT chirality and WSe$_2$ layer number combinations, we observe that the

exciton transfer efficiency can be significantly modulated due to band alignment. The transfer process shows a resonant behavior, leading to a pronounced enhancement in excitation with fast transfer from the WSe$_2$ exciton states. Employing this unique excitation process, we demonstrate simultaneous bright emission from an array of CNTs with varied chiralities and orientations. These findings highlight exciton harvesting using mixed-dimensional heterostructures as a novel approach for overcoming the dimensional limitations in optical processes.

## Results

### Exciton transfer in mixed-dimensional heterostructures

We begin by studying mixed-dimensional heterostructures consisting of a 2D material flake on top of an individual air-suspended CNT. The CNTs are grown over trenches by chemical vapor deposition. For the 2D material, WSe$_2$ is selected owing to its high stability at room temperature and a large bandgap that could offer sufficiently high energy excitons for transferring into CNTs. A WSe$_2$ flake is placed on top of the tubes by utilizing the anthracene-assisted transfer technique[26,27]. The CNT/WSe$_2$ heterostructures are therefore entirely free-standing as depicted in Fig. 1a, b, which eliminates the substrate-induced inhomogeneity and dielectric screening effects[28,29]. The morphology of the heterostructure is examined with an atomic force microscope (Supplementary Fig. 1), and an intimate contact between the CNT and the WSe$_2$ is confirmed.

Photoluminescence excitation (PLE) spectroscopy can provide evidence for the exciton transfer process. We first focus on a CNT with chirality $(n, m) = (9, 8)$ as an example. Figure 1c shows a PLE map from a pristine suspended $(9, 8)$ CNT before the transfer, with single main peaks for each of the excitation and emission energies corresponding to $E_{22}$ and $E_{11}$ transitions, respectively[12]. After transferring a monolayer WSe$_2$ flake, PL is still bright with 77% of the intensity before the transfer (Fig. 1d). Both $E_{11}$ and $E_{22}$ peaks show redshifts of 33 and 54 meV,

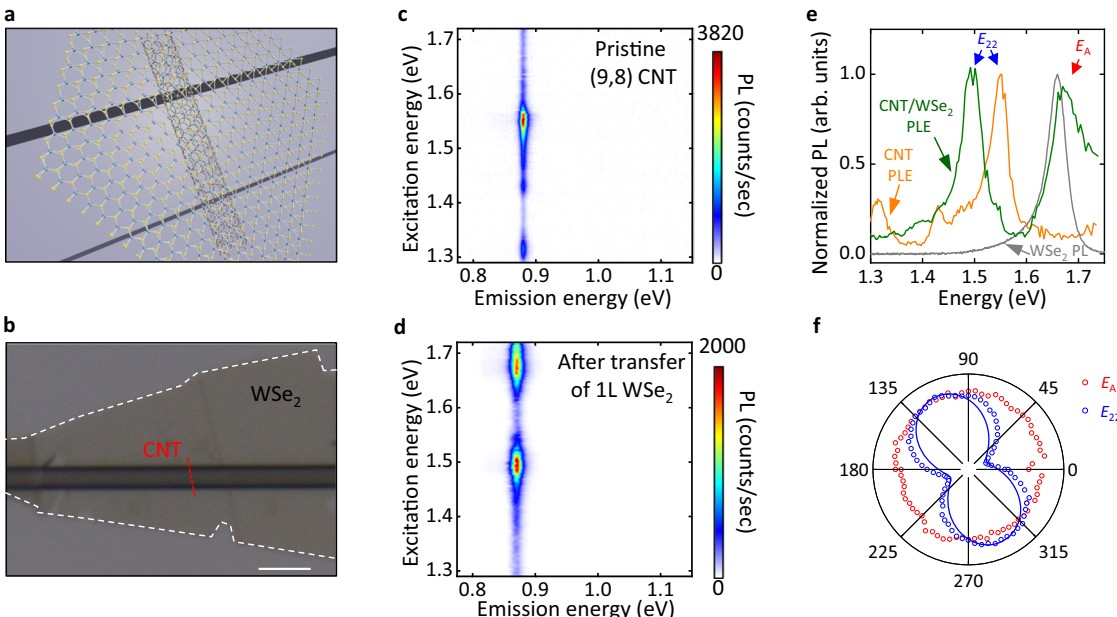

**Fig. 1 | Signature of exciton transfer in a 1D-2D heterostructure. a** A schematic of a suspended CNT/WSe$_2$ heterostructure. **b** An optical microscope image of the $(9,8)$ CNT/1L WSe$_2$ heterostructure. The substrate is SiO$_2$/Si. The scale bar is 5 $\mu$m. **c, d** PLE maps of the $(9,8)$ CNT before (**c**) and after (**d**) the transfer of the 1L WSe$_2$. The excitation power is 10 $\mu$W and the excitation polarization is aligned to CNT axis. **e** Normalized PLE spectra of integrated $E_{11}$ emission for the pristine $(9,8)$ CNT (orange) and after the transfer of 1L WSe$_2$ (green), and a PL spectrum taken from the

suspended 1L WSe$_2$ region away from the nanotube in the same sample (gray). The PL spectrum is normalized to the $E_A$ peak. **f** Excitation polarization dependence for $E_A$ excitation (1.676 eV, red circles) and $E_{22}$ excitation (1.521 eV, blue circles). PL emission is plotted as a function of angle with respect to the trench, where 0 degrees correspond to the direction along the trench. The excitation power is 10 $\mu$W. The blue line is a fit to a cosine squared function.

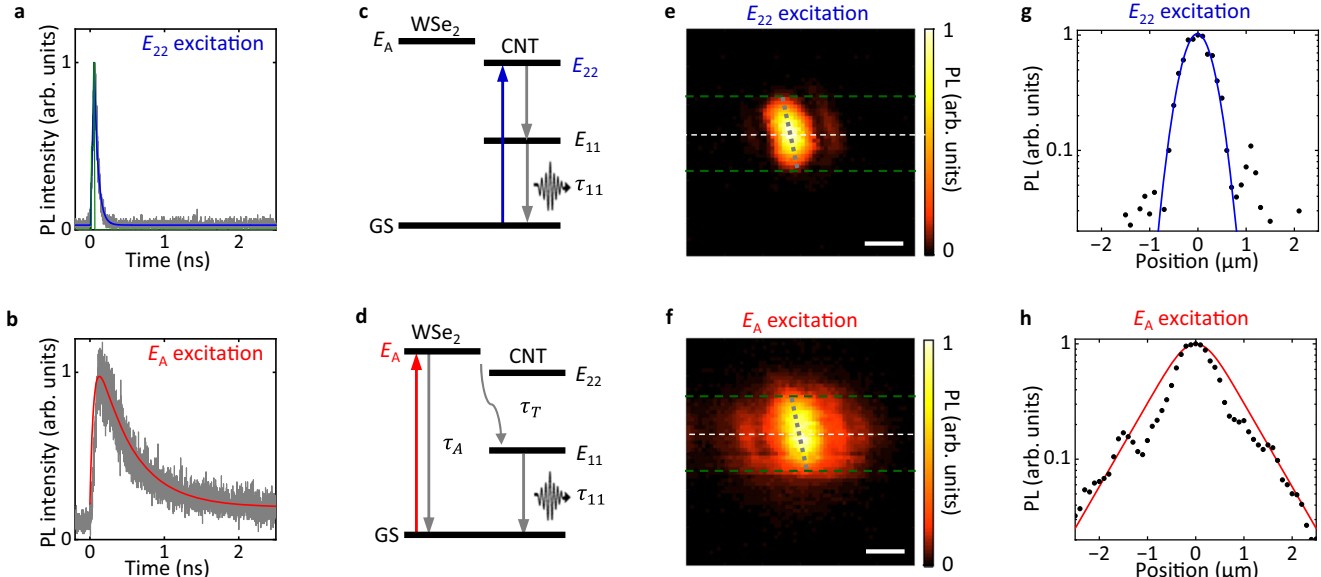

**Fig. 2 | Exciton reservoir effect in 1D-2D heterostructures. a, b** PL decay curves taken from the (9,8) CNT/4L WSe$_2$ heterostructure at $E_{22}$ (1.494 eV, **a**) and $E_A$ excitation (1.653 eV, **b**). Experimental results and IRF are indicated by gray lines and green lines, respectively. Blue and red lines are fits as explained in the text. **c,d**, Energy level diagrams showing the exciton dynamics for $E_{22}$ (**c**) and $E_A$ excitation (**d**). GS indicates the ground state. **e, f** Normalized PL intensity maps from the (9,8) CNT/1L WSe$_2$ heterostructure at $E_{22}$ (1.494 eV, **e**) and $E_A$ excitation (1.653 eV, **f**).

The PL images are constructed by integrating PL emission over a 30-meV-wide spectral window centered at the $E_{11}$ energy. The trench is indicated by the green broken lines and the CNTs are indicated by the broken gray lines. The scale bar is 1 $\mu$m. **g, h** Line profiles from the normalized PL intensity maps indicated by white broken lines in (**e**), (**f**), respectively. Black dots are experimental results and lines are fits. The excitation power is 20 nW for (**a**, **b**) and 10 $\mu$W for (**e**, **f**).

respectively, arising from substantial dielectric screening effect of WSe$_2$[23]. Remarkably, a prominent high-energy peak appears at ~1.673 eV in the excitation dependence.

The origin of the peak can be identified by comparing the PLE spectrum of the heterostructure with the PL spectrum of the suspended WSe$_2$ (Fig. 1e). The PL spectrum (gray curve) shows a single peak from the A excitons, which overlaps with the newly appeared peak in the PLE spectrum of the heterostructure (green curve). The new peak is therefore denoted as $E_A$, indicating a process where absorption by WSe$_2$ A excitons results in emission from CNT $E_{11}$ state. We hypothesize that the exciton transfer process is responsible for coupling the excited and the ground states within different materials. The $E_A$ peak and the $E_{22}$ peak in the PLE spectra (green curve) are comparable in their height, showing that the exciton transfer process is highly efficient despite the different dimensionalities. We also note that the $E_A$ peak in the PLE spectrum shows a considerable tail on the high energy side, which can be explained by the continuum of absorption in WSe$_2$ arising from interband transitions[30].

Excitation polarization measurements offer insights into the dimensionality characteristics of the different excitation pathways. A strong linear polarization dependence emerges from the $E_{22}$ excitation peak as shown in Fig. 1f, consistent with the 1D nature of $E_{22}$ excitons. Conversely, the $E_A$ excitation peak does not display a discernible linear polarization, thereby reflecting the 2D nature of A excitons. This notable polarization distinction enables selective excitation of the samples, either by the A exciton or the $E_{22}$ exciton (Supplementary Fig. 2), thus introducing an additional degree of freedom for excitation manipulation.

Similar exciton transfer processes are observed for heterostructures with thin WSe$_2$ layers, varying from monolayer (1L) to quadlayer (4L). However, this process is substantially less pronounced in WSe$_2$ flakes with a thickness of approximately 10 nm (Supplementary Fig. 3). For subsequent analysis, we therefore concentrate primarily on few-layer WSe$_2$ samples from 1L to 4L.

## Exciton reservoir effect in mixed-dimensional heterostructures

Having identified the exciton transfer process in CNT/WSe$_2$ heterostructures, we carry out time-resolved PL measurements to explore the dynamics of the exciton transfer. A (9,8) CNT/4L WSe$_2$ heterostructure is investigated by comparing the time-resolved PL signals from $E_{11}$ peaks under two distinct excitation conditions.

For $E_{22}$ excitation, a rapid decay curve is observed (Fig. 2a). We extract the decay lifetime as 52 ps using exponential fitting convoluted with the instrument response function (IRF). This small value is consistent with the bright exciton lifetime for suspended (9,8) CNTs[11]. Interestingly, when under $E_A$ excitation, a much slower decay curve is observed (Fig. 2b).

The marked change in the decay time cannot be ascribed to the increase of $E_{11}$ lifetime since no obvious excitation dependence in $E_{11}$ excitons is observed in terms of spectral peak energy and linewidth. Instead, it can be understood by considering the exciton transfer dynamics, which include WSe$_2$ A exciton generation, recombination, and transfer processes (Fig. 2c, d).

In the case of $E_{22}$ excitation, given that the relaxation time from $E_{22}$ to $E_{11}$ excitons is on the order of femtoseconds[31], the rate equation for $E_{11}$ exciton population $n_{11}$ is expressed as

$$\frac{dn_{11}}{dt} = G - \frac{n_{11}}{\tau_{11}}, \qquad (1)$$

where $G$ represents generation rate by the laser, and $\tau_{11} = 52$ ps is the lifetime of $E_{11}$ excitons. When excited at the $E_A$ peak, the rate equation for $n_{11}$ is expressed as

$$\frac{dn_{11}}{dt} = \frac{Qn_A}{\tau_T} - \frac{n_{11}}{\tau_{11}}, \qquad (2)$$

where $Q$ is the fraction of $A$ excitons interacting with the CNT relative to the total $A$ exciton population $n_A$, and $\tau_T$ represents the transfer time. It is worth noting that the rise time in Fig. 2b is as short as that in

Fig. 2a and below the resolution limited by IRF, indicating that the transfer is a fast process.

The rate equation for $n_A$ in the WSe$_2$ flake can be represented as

$$\frac{\mathrm{d}n_A}{\mathrm{d}t} = G - \frac{(1-Q)n_A}{\tau_A} - \frac{Qn_A}{\tau_T}, \quad (3)$$

where $\tau_A$ is the $A$ exciton lifetime. In Eq. (3), we consider $Q$ to be negligibly small as only a small portion of $A$ excitons are transferred due to the higher dimensionality of WSe$_2$. The solution to Eq. (3) is therefore proportional to $\exp(-t/\tau_A)$, and we can accurately reproduce the time-resolved PL in Fig. 2b by setting $\tau_A = 500$ ps and solving Eq. (2). The A exciton lifetime extracted is one order of magnitude larger than that in CNTs (52 ps) as expected from the smaller binding energy, and consistent with previously reported values from mechanically exfoliated WSe$_2$[32]. Essentially, the WSe$_2$ flake serves as an exciton reservoir, continuously supplying excitons to CNTs which leads to the slow decay curve. We note that a smaller $\tau_A$ of ~200 ps is observed in the case of a thinner WSe$_2$ bilayer (Supplementary Fig. 4).

The exciton reservoir effect is also apparent in imaging measurements. We use a three-dimensional motorized stage to scan the (9,8) CNT/1L WSe$_2$ sample shown in Fig. 1 for obtaining PL images. Since the collection spot is much larger than the excitation spot, emission outside the excitation laser is also collected while the image resolution is determined by the laser spot size. The images therefore represent excitation efficiency profiles for CNT PL, which can be used to identify non-local excitation processes. Figure 2e displays a PL excitation image with the excitation at the $E_{22}$ peak, presenting a typical suspended CNT image with the length of the bright PL area constrained by the trench and the width determined by the Gaussian laser profile. Notably, the PL excitation image for $E_A$ excitation appears spatially expanded, as shown in Fig. 2f. This enlarged PL image indicates that A excitons excited at a distance also funnel into the CNT after diffusion. The WSe$_2$ flake therefore also acts as a spatial reservoir for exciting CNTs. We also note that the image enlargement occurs not only in the suspended region but also on the substrate, indicating that A excitons are more resilient to substrate effects than $E_{11}$ excitons.

Line profiles of the PL images along the trench for $E_{22}$ and $E_A$ excitation are illustrated in Fig. 2g and h, respectively. The $E_{22}$-excited profile is fitted using a Gaussian function, yielding a $1/e^2$ radius of the laser spot $r = 0.58$ $\mu$m. For the $E_A$ excitation, the PL intensity is proportional to the density of A exciton transferred to the CNT position, assuming that the exciton transfer is a linear process. The line profile at $E_A$ excitation is hence fitted to a solution of a steady-state one-dimensional diffusion equation

$$D\frac{\mathrm{d}^2 n_A(x)}{\mathrm{d}x^2} - \frac{n_A(x)}{\tau_A} + \frac{G}{\sqrt{2\pi r^2}}e^{-2x^2/r^2} = 0, \quad (4)$$

where $x$ is the position along the trench direction and $D$ is the diffusion coefficient. Given $r = 0.58$ $\mu$m, the simulated profile with a diffusion length $L = \sqrt{D\tau_A} = 0.60$ $\mu$m matches the experimental data as shown in Fig. 2h. The diffusion lengths are also estimated for heterostructures with thicker WSe$_2$ flakes, reaching up to 1.10 $\mu$m (Supplementary Fig. 5). It is worth mentioning that the diffusion length here is larger than previously reported values[33], possibly due to the suspended structure of the WSe$_2$ flake which has reduced scattering sites.

## Band alignment tuning and resonant exciton transfer

The efficient exciton reservoir effect observed in both spatial and temporal domains suggests the possibility of a greatly enhanced exciton transfer under suitable band alignment. To explore the optimized structure, we take advantage of the variable bandgap of CNT with chirality. We have fabricated 34 CNT/WSe$_2$ heterostructures using

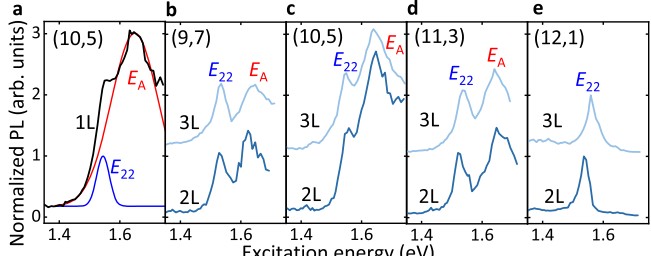

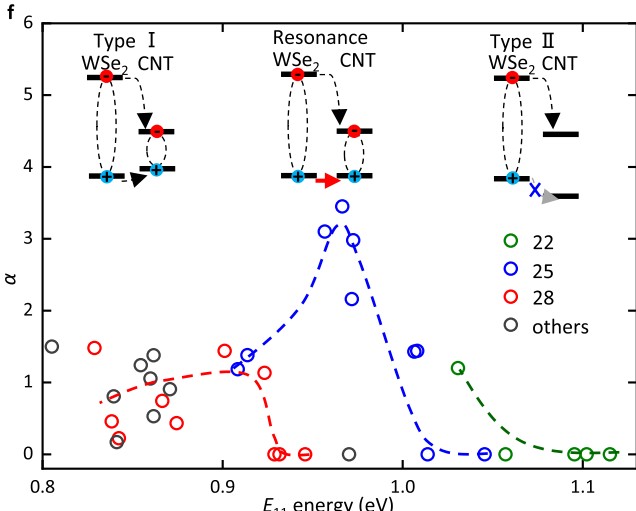

**Fig. 3 | Band alignment tuning and resonant exciton transfer. a–e** Normalized PLE spectra of integrated $E_{11}$ emission for the (10,5) CNT/1L WSe$_2$ sample (**a**), (9,7) CNT/2L and 3L WSe$_2$ samples (**b**), (10,5) CNT/2L and 3L WSe$_2$ samples (**c**), (11,3) CNT/2L and 3L WSe$_2$ samples (**d**), (12,1) CNT/2L and 3L WSe$_2$ samples (**e**). The PLE spectra are normalized to the $E_{22}$ peak intensity. In (**b–e**), baselines of PLE spectra for the samples with 2L and 3L WSe$_2$ are 0 and 1, respectively. The excitation power is 10 $\mu$W. **f** Factor $\alpha$ as a function of $E_{11}$ energy from all samples. CNT families of 22, 25, 28, and others are indicated by green, blue, red, and gray circles, respectively. Broken lines are guide to the eye. (Inset) Schematics of band alignment of CNT/WSe$_2$ heterostructures. With varying CNT chirality, the band alignment transition from type-I to type-II occurs. Black arrows indicate the exciton tunneling process, while a red arrow indicates the resonant tunneling of holes when the valence bands match.

17 different chiralities (see Supplementary Table 1 for the list of samples) and have quantitatively characterized the exciton transfer process.

Figure 3a displays the PLE spectrum from a (10,5) CNT/1L WSe$_2$ heterostructure, which has demonstrated the largest enhancement of the exciton transfer. The $E_A$ peak is dominating the spectrum, and the $E_{22}$ peak is almost overwhelmed. Gaussian peak fits are carried out to distinguish the two excitation peaks. The efficiency factor $\alpha = I(E_A)/I(E_{22})$ is defined to quantify the $E_A$ excitation process with respect to the $E_{22}$ excitation, where $I(E_A)$ and $I(E_{22})$ are the peak intensities of $E_A$ and $E_{22}$, respectively. In this heterostructure, the exciton population is multiplied by $\alpha = 3.5$ compared with $E_{22}$ excitation. Under such an effective excitation through A excitons, we note that the exciton-exciton annihilation (EEA) process[12] is strong and the $\alpha$ factor is underestimated. A PLE measurement at a low laser power to avoid the EEA effect leads to an increased $\alpha$ factor of 6.2 (see Supplementary Fig. 6). From the $\alpha$ factor, we estimate that approximately 3.8% of the A excitons transfer into the nanotube, taking into account the different absorption areas and absorption coefficients (see Supplementary Note 8). While this value may not seem high, it is actually substantial because the exciton population in WSe$_2$ is orders of magnitude larger than in CNTs.

To elucidate the origin of the enhanced transfer, we plot the PLE spectra for the heterostructures with bilayer (2L) and trilayer (3L) WSe$_2$ transferred onto CNTs with varying chiralities of (9,7), (10,5), (11,3), and (12,1) in Fig. 3b–e, respectively. The selected chiralities here belong to the same family ($2n + m = 25$). The PLE spectra from (9,7) and (11,3) samples exhibit clear $E_A$ peaks, although weaker than that in (10,5) CNTs. In contrast, the $E_A$ peak is unresolved in (12,1) samples, indicating a strong suppression of the exciton transfer process.

The observed chirality-dependent exciton transfer processes cannot be explained by the commonly observed Förster-type energy transfer in 2D heterostructures[34,35]. For the energy transfer process where WSe$_2$ A exciton is the donor and CNT $E_{22}$ exciton is the acceptor, we do not expect changes in the transfer efficiency unless the spectral overlap is modulated substantially. The chiralities shown in Fig. 3b–e have similar $E_{22}$ energies but exhibit large changes in the efficiency factors, inconsistent with Förster-type energy transfer.

Instead, we interpret our data with a direct exciton tunneling process, which should be sensitive to the band alignment between the two materials[36,37]. For CNT/2L WSe$_2$ samples, the $E_{11}$ energies for chiralities of (9,7), (10,5), (11,3), and (12,1) are 0.914, 0.973, 1.008, and 1.046 eV, respectively. Since the $E_{11}$ energy reflects the bandgap of CNTs, the band alignment should be varied, as illustrated in Fig. 3f. For chiralities with smaller bandgap, type-I heterojunction is expected where exciton tunneling is allowed. As the bandgap gets larger, the valence band offset between CNT and WSe$_2$ becomes smaller. When the bands match, we expect resonant tunneling for the holes, and this maybe the case for (10,5) CNTs with significantly enhanced $E_A$ excitation peak. With further increase in the bandgap, transition to type-II heterojunction should occur. The exciton transfer would then become inhibited due to the barrier for the hole tunneling, which is consistent with the PLE spectra for (12,1).

We perform density functional theory studies to examine the band alignment within the heterostructure (Supplementary Fig. 7). Results show that offsets in the valence band is less significant than that in the conduction band. The variation of chirality from lower $E_{11}$ energy to a higher one ultimately instigates the reversal of the valence band offset, thus leading to the transition from type-I to type-II. The

theoretical findings corroborate with the empirical results described above, supporting the picture that band alignment transition is taking place and confirming the exciton transfer process.

The transition is more clearly observed by plotting the $\alpha$ factor against the $E_{11}$ energy, as shown in Fig. 3f. For family number 25, the resonant peak can be seen at $E_{11}$ energy around 0.97 eV. The band alignment transitions are consistently observed for families 22 and 28 with the threshold $E_{11}$ energy of the transition increasing with the family number. The CNT families, recognized for impacting the band structure of CNTs via trigonal warping and curvature effects[38], may also be influencing the band alignment in the heterostructures here. It should be noted that there is no observable resonance in families 22 and 28, which could be attributed to the absence of a chirality with an exactly matched band energy.

The highly efficient exciton transfer at the resonance suggests it is a rapid process, and we perform Monte Carlo simulations to gain more insight. Exciton generation, diffusion, transfer, and recombination processes are included, and we find the transfer time $\tau_T = 1.1$ ps reproduce the experimental value of $\alpha = 6.2$ (see Supplementary Note 8). In contrast, mixed-dimensional heterostructures consisting of zero-dimensional quantum dots and 2D TMDs show a long exciton transfer time in the range of 1–10 ns[39,40]. Such a long transfer time is primarily caused by complex interfaces consisting of shells and ligands. The transfer process observed in this study is remarkably fast and comparable with homogeneous dimensional structures, such as 2D-2D TMD and 1D-1D CNT heterostructures[34,41]. The rapid transfer process can be ascribed to the intimate contact between the CNT and WSe$_2$, facilitated by the uniform van der Waals interface and the presence of a resonant tunneling condition.

## Overcoming the dimensional restrictions on excitation in a CNT array/WSe$_2$ heterostructure

With relaxed optical selection rules and pronounced exciton reservoir effects, the A exciton based excitation has a more distinct effect on an array of CNTs. The array under investigation comprises of four tubes located from left to right with chiralities of (8,7), (9,8), (10,5), and (10,5), and is covered by a monolayer flake. Figure 4a shows a PL

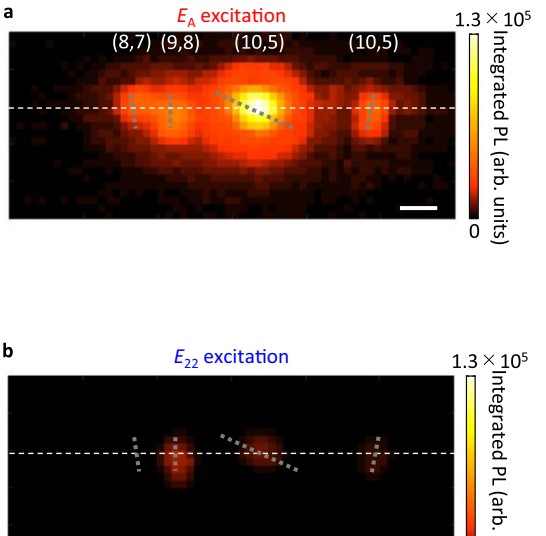

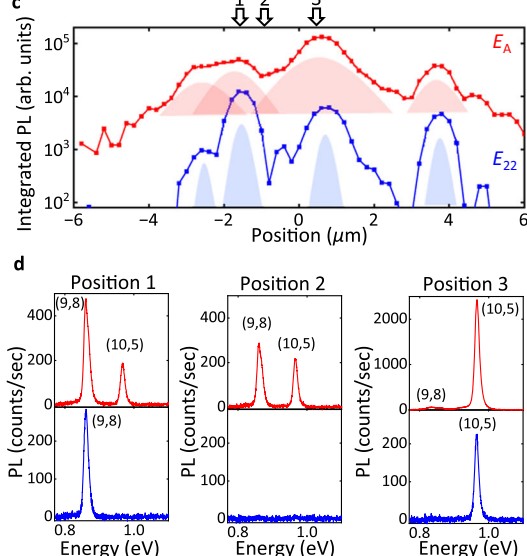

**Fig. 4 | Simultaneous excitation of a CNT array in a mixed-dimensional heterostructure. a, b** PL intensity maps for $E_A$ (1.664 eV, **a**) and $E_{22}$ (1.494 eV, **b**) excitation. The PL images are constructed by integrating the full PL spectrum. The CNT chiralities are (8,7), (9,8), (10,5), and (10,5) from left to right, and the positions and orientations are indicated by the broken gray lines. The scale bar is 1 µm. **c** Line

profiles indicated by white broken lines in (**a**, red) and (**b**, blue). **d** PL spectra taken at Positions 1, 2, and 3 as indicated in (**c**). The upper three PL spectra are taken with $E_A$ excitation and the bottom three PL spectra are taken with $E_{22}$ excitation. The excitation power is 3 µW.

excitation image with $E_A$ excitation, whereas Fig. 4b is an excitation image taken with $E_{22}$ excitation for the (9,8) tube. The excitation at the $E_A$ energy results in high intensity PL from all tubes and a quasi-2D excitation image spanning the trench. We note that the two (10,5) CNTs exhibit different PL intensities, which is primarily due to their different lengths[12]. The line profile (Fig. 4c, red) shows spatially continuous emission over 10 $\mu$m, resulting from overlapping tube images enlarged by exciton diffusion. In comparison, tube images are completely isolated in Fig. 4b and its line profile (Fig. 4c, blue). Furthermore, the (8,7) tube is unnoticeable since the excitation energy is detuned from its $E_{22}$ energy and the polarization angle is misaligned. In general, achieving simultaneous excitation across all tubes remains a challenge due to different $E_{22}$ energies and varying angles amongst the CNTs.

The diffusion process can be over one micron (Supplementary Fig. 5), allowing for enhanced versatility in exciting multiple tubes. In Fig. 4d, we plot PL spectra at three representative positions for $E_A$ excitation (red) and $E_{22}$ excitation (blue). Simultaneous PL emission from spatially distant (9,8) and (10,5) tubes is demonstrated by $E_A$ excitation at the region in between (Position 2, red). In contrast, no PL is observed under $E_{22}$ excitation at this position (blue).

## Discussion

In conclusion, an efficient exciton transfer process within CNT/WSe$_2$ heterostructures has been observed. Excitation via WSe$_2$ A excitons presents several distinctive features compared to traditional $E_{22}$ excitation: larger absorption areas, broader spectra, and tolerance to polarization misalignment. Through the examination of 17 different CNT chiralities, it has been confirmed that the exciton transfer is significantly modulated by the band alignment. The (10,5) CNT/WSe$_2$ heterostructure demonstrates the resonant exciton transfer, leading to a considerable enhancement of $E_A$ excitation efficiency and a rapid transfer time of 1.1 ps. The ability to simultaneously illuminate CNT arrays of varying chiralities and orientations, resulting in a quasi-2D image under $E_A$ excitation, is also demonstrated. These observations propose that the exciton transfer process in mixed-dimensional heterostructures could be harnessed for generation of excitons in CNTs, thus providing a promising avenue to surpass the excitation constraints in low-dimensional materials. The high tunability of the band alignment indicates the potential for more intriguing excitons to emerge in this system, for example, indirect excitonic states at the mixed-dimensional interface under type-II band alignment.

## Methods

### Air-suspended carbon nanotube growth

In preparing air-suspended CNTs, we utilize silicon dioxide (SiO$_2$)/silicon (Si) substrates with trenches[12]. The process begins with patterning of alignment markers and trenches onto the Si substrates, utilizing electron-beam lithography. The trenches, with lengths of approximately 900 $\mu$m and widths varying between 0.5 and 3.0 $\mu$m, are subsequently formed by dry etching. The substrate is then subjected to thermal oxidation, resulting in the formation of a SiO$_2$ film inside the trenches, with a thickness typically ranging from 60 to 70 nm. The next step involves a further instance of electron-beam lithography to designate catalyst areas along the trench edges. An iron (Fe) film with a thickness of approximately 1.5 Å is then deposited by an electron beam evaporator, serving as a catalyst in the CNT growth. The final step of the process is the synthesis of CNTs, achieved through alcohol chemical vapor deposition at a temperature of 800 °C for 1 min. Optimization of the iron film thickness is crucial in controlling the yield for predominantly producing isolated synthesized CNTs. PLE spectroscopy is conducted to identify the chiralities of suspended CNTs by comparing $E_{11}$ and $E_{22}$ energies to tabulated data[12]. In order to form the heterostructures with WSe$_2$, we specifically select isolated, fully suspended chirality-identified CNTs with lengths in the range of 0.5 to 2.0 $\mu$m.

### Anthracene crystal growth

For the purpose of stamping WSe$_2$ flakes onto CNTs, anthracene crystals are grown through an in-air sublimation process[26,27]. The procedure commences with the placement of anthracene powder onto a glass slide, which is subsequently heated to 80 °C. A secondary glass slide is positioned above the anthracene source, generally maintaining a 1 mm gap. Following this setup, thin, large-area single crystals begin to grow out of the glass surface. To enhance the growth of thin, large-area crystals, the glass slides are patterned using ink from commercially available markers, effectively inhibiting the nucleation of three-dimensional crystals. The typical growth time for anthracene crystals is 10 h.

### WSe$_2$ transfer using anthracene crystals

WSe$_2$ crystals are purchased from HQ graphene. Flakes of WSe$_2$ are prepared on conventional 90-nm-thick SiO$_2$/Si substrates by employing a mechanical exfoliation technique. Layer number is determined by optical contrast. A polydimethylsiloxane (PDMS) sheet supported by glass is used to collect a single anthracene crystal, creating an anthracene/PDMS stamp. The targeted WSe$_2$ flakes on the substrate are subsequently picked up by pressing the anthracene/PDMS stamp. Rapid separation of the stamp (>10 mm/s) ensures the adherence of the anthracene crystal to the PDMS sheet. The stamp is then applied to the receiving substrate with the desired chirality-identified CNT, whose position is determined by a prior measurement. Precise position alignment is accomplished with the aid of markers prepared on the substrate. Gradual retraction of the PDMS (<0.2 $\mu$m/s) allows for the anthracene crystal carrying the WSe$_2$ flake to settle on the receiving substrate. The anthracene crystal is then removed via sublimation in air at 110 °C over a 10-min period, leaving a clean suspended CNT/WSe$_2$ heterostructure. This completely dry process eliminates the risk of contamination from solvent. Moreover, the solid single-crystal anthracene serves to shield the 2D flakes and the CNT throughout the transfer, ensuring that the CNT/WSe$_2$ heterostructure experiences minimal strain[26,27]. We note that some of the CNT/WSe$_2$ samples exhibit substantially small $E_{11}$ and $E_{22}$ energy shifts of less than 5 meV, suggesting a lack of contact between the CNT and the WSe$_2$. These samples are excluded in the following study.

### Photoluminescence measurements

A custom-built confocal microscopy apparatus is utilized to perform PL measurements for CNT $E_{11}$ excitons at ambient temperature within an atmosphere of dry nitrogen gas[12,23]. A variable wavelength Ti:sapphire laser serves as a continuous-wave excitation source, with its power controlled via neutral density filters. The laser beam is focused onto the samples with an objective lens featuring a numerical aperture of 0.65 and a working distance of 4.5 mm. The $1/e^2$ diameter of the laser spot is 1.16 $\mu$m. The confocal pinhole defines the collection spot size, which is approximately 5.4 $\mu$m in diameter. A longpass filter with a cut-on wavelength of 1050 nm is used to collect the PL emission from CNTs. PL is gathered through the same objective lens and detected by a liquid-nitrogen-cooled 1024-pixel indium gallium arsenide diode array attached to a spectrometer. A 150-lines/mm grating is used to obtain a dispersion of 0.52 nm/pixel at a wavelength of 1340 nm. PL excitation images are taken by using a three-dimensional motorized stage to scan the CNT/WSe$_2$ samples in our confocal microscopy system. A PL spectrum is collected at each position, and PL excitation images are constructed by integrating the CNT emission. For photoluminescence measurements of WSe$_2$ A excitons, a 532-nm laser and a charge-coupled device camera are employed. Polarization is aligned to the tube unless otherwise noted.

## Time-resolved measurements

Approximately 100 femtosecond pulses at a repetition rate of 76 MHz from a Ti:sapphire laser is utilized for excitation. The laser beam is directed onto the sample using an objective lens with a numerical aperture of 0.85 and a working distance of 1.48 mm. The PL from the center of the nanotube within the heterostructure is coupled to a superconducting single-photon detector with an optical fiber, and a time-correlated single-photon counting module is used to collect the data. IRFs dependent on the detection wavelength are acquired by dispersing supercontinuum white light pulses with a spectrometer. The experiments are conducted at room temperature in air.

## Data availability

All the data generated in this study have been deposited in the R2DMS-GakuNinRDM database at https://dmsgrdm.riken.jp/sdgab/. Source data are provided with this paper.

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

## Acknowledgements

Parts of this study are supported by JSPS (KAKENHI JP22K14624 to D.Y., JP22K14625 to S.F., JP21K14484 to M.M., JP22K14623 to C.F.F., JP21H05233 to S.O., JP22F22350 to Y.R.C., JP23H00262, JP20H02558 to Y.K.K.) and MEXT (ARIM JPMXP1222UT1135). Y.R.C. is supported by JSPS (International Research Fellow). N.F. and C.F.F. are supported by the RIKEN Special Postdoctoral Researcher Program. We thank the Advanced Manufacturing Support Team at RIKEN for technical assistance.

## Author contributions

N.F. carried out sample preparation and performed measurements on all the samples. Y.R.C. performed atomic force microscope measurements and assisted in other measurements. D.Y. and S.F. contributed to the time-resolved PL measurements. M. M., Y.G., and S.O. performed density functional theory calculations. Y.R.C., C.F.F., and K.N. assisted in sample preparation. K.O. aided in the development of the anthracene-assisted transfer method. Y.K.K. supervised the project. N.F. and Y.K.K. co-wrote the manuscript, with all authors providing input and comments on the manuscript.

## Competing interests

The authors declare no competing interests.
