## [Peer Review File · Nature Communications]

Reviewers' Comments:

Reviewer #1:

Remarks to the Author:

In this paper, the author reported an exciton transfer process in CNT/WSe₂ mixed-dimensional heterostructures and the exciton transfer process is significantly modulated by the band alignment. Moreover, the author demonstrates simultaneous bright emission from an array of CNTs with varied chiralities and orientations.

This is an interesting and nice work. However the manuscript needs to be improved. I would suggest this paper could be accepted after minor revision. Some comments are reported below:

(1) The author is suggested to mark the CNT position and orientation in Figure 2e,f and Figure 4a,b.

(2) As shown in Figure 2e,f and Figure 4a,b, the emission area is spatially extended under EA excitation. The Author gives explanation for this phenomena on Page 8 line 160 as "This enlarged PL image indicates that A excitons excited at a distance also funnel into the CNT after diffusion". This is not right because the PL image is showing the photons emitted from the CNT rather WSe₂, A excitons in WSe₂ at a distance transport and funnel into the CNT should not make the emission area of the CNT spatially extended. The author is suggested to give a deeper analysis and discussion here.

Reviewer #2:

Remarks to the Author:

Dear Editor,

The manuscript reports on exciton transfer in CNT/WSe₂-ML heterostructures. The authors claimed that excitons with longer-lifetime in WSe₂ ML are funneled into CNT through diffusion. When the band alignment is resonant, they observe substantially more efficient excitation of CNTs via WSe₂ compared to direct excitation of the nanotube, and they demonstrate simultaneous bright emission from an array of CNTs with varied chiralities and orientations.

In Fig. 1e, the authors should show the emission of the CNT without the WSe₂ ML. The authors used PLE polarization dependence of EA and E22 peaks in Fig. 1f to conclude (in my opinion at this stage in the manuscript they should hypothesize the idea instead of concluding) that exciton transfer took place. This can not be conclusive if they never did similar measurements for the CNT without the WSe₂ ML

In Line 96, the authors stated "Remarkably, a prominent high-energy peak appears at ~1.673 eV in the 97 excitation dependence, which is a signature of the exciton transfer process". This is a speculation because based on Fig. 1e, this peak corresponds to WSe₂ ML emission. Yes, in the presence of CNT, the peak is broader on the high energy side, which can be due to many reasons other than exciton transfer (e.g. technical issues because the authors did not show the emission data at photon energies beyond 1.75 eV, dielectric screening effect in the presence of CNT, etc.) On Lines 103-105, the authors stated that "The EA peak and the E22 peak in the PLE spectra are comparable in their height, showing that the exciton transfer process is highly efficient despite the different 105 dimensionalities". Again, this is a speculation because the fact that the intensity of the peak remained almost the same in the presence of the CNT indicates that there is no charge transfer nor exciton transfer. To prove this, the authors should consider the fact that the overlap area (CNT/WSe₂) from which the emission originates is much smaller than the excitation spot size, thus, if we assume that there is exciton transfer, the emission of these transferred excitons would be much weaker than the emission of WSe₂ excitons (in the absence of CNT).

In my opinion, based on Fig. 1c and 1d, it is clear that there is energy transfer from WSe₂ ML to CNT because the emission of the CNT (around 0.88 eV) increased in the presence of the WSe₂ ML. In my opinion, although the studied system is very interesting, the experiments are well designed, and the results are well presented, the manuscript should not be accepted in Nature Communications because the conclusions are speculative and not proved. If there are convincing results to conclude exciton transfer, I would recommend publication in Nature Comm, but the results suggest that what took place at this interface is energy transfer, which is not new taking

into account that previous reports have already observed energy, and/or charge transfer at the interface of 2D-TMDs and other nanostructures such as Plasmonics, QDs, and 2D/2D.

Reviewer #3:

Remarks to the Author:

In this work, the authors reported the exciton transfer from two-dimensional layered WSe₂ to one-dimensional carbon nanotubes in the mixed-dimensional heterostructures. They constructed the mixed-dimensional heterostructure by transferring WSe₂ of few layers to CNTs that were grown across trenches, and measured the photoluminescence excitation (PLE) spectroscopy of the heterostructures. From time-resolved PL measurements, they found that the decay lifetime is much longer for suspended (9, 8) CNTs/WSe₂ excited by EA than that by E22. The PL intensity maps also showed that the PL is spatially expanded if excited by EA than E22, and a diffusion length of excitons of 0.6 μm was estimated. The resonant exciton transfer was further studied by comparing heterostructures with CNTs of different chirality. DFT calculations of the energy band alignment were performed to explain the enhancement and suppression of exciton transfer for different CNTs. Although the exciton transfer process within CNT/WSe₂ heterostructures reported in this work is new observation, the reviewer can not recommend publication of this work in Nature Communications due to the following reasons, and possible revisions are also suggested below.

1. The authors measured the CNT/WSe₂ mixed dimensional heterostructures of different CNTs. However, more experimental details should be provided.
2. How did the authors assign the chirality of the suspended CNTs?
3. How did the authors ensure the contact between CNTs and WSe₂? Particularly, CNTs are across trenches, and it is natural that CNTs will bend down toward the trench. If they compare CNTs across trenches of different width, it is expectable that different results will be obtained.
4. The authors should provide morphology data, such as SEM or AFM, where CNTs are visible.
5. Raman spectroscopy could be possible to confirm the interaction between CNTs and WSe₂?
6. Did the authors compare the PLE of the heterostructure for suspended CNTs and supported CNTs on the substrate and what is the difference?
7. For the PL maps shown in Figure 2 and 4, what is the wavelength range for the signal was collected? Is it PL from WSe₂ or CNTs, or both?
8. What the excitation wavelength for the PL of WSe₂ shown in Figure 1e? Do they expect PL intensity of WSe₂ using EA excitation?
9. The authors used a collection spot size of 5.4 μm in diameter as they described in the experimental section, how could the CNTs in Figure 4a and 4b be resolved spatially?
10. Two (10, 5) CNTs are shown in Figure 4a, but the intensities are apparently different, can the authors explain the difference?

Response to reviewer reports for manuscript No. NCOMMS-23-39012-T by Fang *et al.*

We thank the reviewers for taking their time to review the manuscript. We are grateful for the helpful comments which we have used to improve our manuscript. Our point-by-point response is provided below.

Response to Reviewer #1:

In this paper, the author reported an exciton transfer process in CNT/WSe₂ mixed-dimensional heterostructures and the exciton transfer process is significantly modulated by the band alignment. Moreover, the author demonstrates simultaneous bright emission from an array of CNTs with varied chiralities and orientations.

This is an interesting and nice work. However, the manuscript needs to be improved. I would suggest this paper could be accepted after minor revision. Some comments are reported below:

We thank the reviewer for the compliment, and we are very happy to hear that the manuscript could be accepted after minor revision. Below, we have addressed the comments point by point.

1. The author is suggested to mark the CNT position and orientation in Figure 2e,f and Figure 4a,b.

As suggested, we have revised Fig. 2e,f and 4a,b to mark the CNT position and orientation.

2. As shown in Figure 2e,f and Figure 4a,b, the emission area is spatially extended under EA excitation. The Author gives explanation for this phenomena on Page 8 line 160 as “This enlarged PL image indicates that A excitons excited at a distance also funnel into the CNT after diffusion”. This is not right because the PL image is showing the photons emitted from the CNT rather WSe₂, A excitons in WSe₂ at a distance transport and funnel into the CNT should not make the emission area of the CNT spatially extended. The author is suggested to give a deeper analysis and discussion here.

We sincerely thank the reviewer for pointing this out. We realized that the explanation for the PL images was insufficient. The PL images shown in Fig. 2e,f and Fig. 4a,b are not PL “emission” images but PL “excitation” images. The PL images are constructed by collecting a PL spectrum at each position and integrating the emission from CNT. Since the collection spot diameter of 5.4 μm is much larger than the laser spot diameter of 1.2 μm , emission outside the excitation laser spot is also detected. Therefore, the images represent excitation efficiency profiles for CNT PL where the image resolution is determined by the laser spot size.

The PL excitation images can be used to identify non-local excitation processes. The enlarged PL excitation image indicates that the laser spot far away from the CNT could still contribute to the PL emission from the CNT. This is explained in terms of diffusion of the WSe₂ A excitons excited at a distance followed by transfer into the CNT (Page 8, second and third paragraphs).

To clarify this point, we have added a description of the PL excitation imaging measurement in the second paragraph of Page 8:

“Since the collection spot is much larger than the excitation spot, emission outside the excitation laser is also collected while the image resolution is determined by the laser spot size. The images therefore represent excitation efficiency profiles for CNT PL, which can be used to identify non-local excitation processes.”

We have also added the laser spot size and the following description of the PL imaging in the paragraph for photoluminescence measurements in Methods section:

“PL excitation images are taken by using a three-dimensional motorized stage to scan the CNT/WSe₂ samples in our confocal microscopy system. A PL spectrum is collected at each position, and PL excitation images are constructed by integrating the CNT emission.”

To emphasize that Fig. 2e,f and Fig. 4a,b are PL excitation images, we have also changed “PL images” to “PL excitation images” in Pages 8 and 13.

Response to Reviewer #2:

The manuscript reports on exciton transfer in CNT/WSe₂-ML heterostructures. The authors claimed that excitons with longer-lifetime in WSe₂ ML are funneled into CNT through diffusion. When the band alignment is resonant, they observe substantially more efficient excitation of CNTs via WSe₂ compared to direct excitation of the nanotube, and they demonstrate simultaneous bright emission from an array of CNTs with varied chiralities and orientations.

We sincerely thank the reviewer for concisely summarizing the results. Before the detailed point-by-point response, we would like to address the overall criticism by the reviewer that the experimental observation should be interpreted as energy transfer instead of exciton transfer. We agree this is an important point because both energy transfer and exciton transfer will result in similar PLE spectra, and we are very grateful that the reviewer provided this perspective. We agree that we should take a more conservative view on this matter, but we would like to bring to your attention that we do explain the reason why Forster-type energy transfer can be ruled out. In the section “Band alignment tuning and resonant exciton transfer” starting from Page 9, we discuss the bandgap dependent transfer process showing a resonant behavior, as the reviewer also mentions above. On Page 10, Lines 214-219, we explicitly rule out the Forster-type energy transfer based on this resonant behavior:

“For energy transfer process where WSe₂ A exciton is the donor and CNT E₂₂ exciton is the acceptor, we do not expect changes in the transfer efficiency unless the spectral overlap is modulated substantially. The chiralities shown in Fig. 3b-e have similar E₂₂ energies but exhibit large changes in the efficiency factors, inconsistent with Forster-type energy transfer.”

We therefore interpret the chirality-dependent phenomenon with the exciton transfer process, which is modulated by the band alignment. The validity of the model is further examined by the density functional theory as shown in Supplementary Note 7. The discussion can be found in the third paragraph of Page 11.

Since the discussion regarding the resonant behavior starts later in the manuscript, we agree that we should take a more conservative standpoint in the earlier sections. We have made changes to the main text accordingly, as detailed in the point-by-point response below.

In Fig. 1e, the authors should show the emission of the CNT without the WSe₂ ML.

As suggested, we have revised Fig. 1e to include the PLE spectrum of the CNT before the transfer of the WSe₂ flake.

The authors used PLE polarization dependence of EA and E22 peaks in Fig. 1f to conclude (in my opinion at this stage in the manuscript they should hypothesize the idea instead of concluding) that exciton transfer took place. This can not be conclusive if they never did similar measurements for the CNT without the WSe₂ ML.

We agree that we should take a more conservative standpoint at this stage, as energy transfer could also result in a similar PLE map and polarization dependence. We revised the text such that exciton transfer is hypothesized first to avoid drawing any conclusions at this point. The detailed changes are shown below:

On Page 5, the second paragraph, we have deleted the clause “which is a signature of the exciton transfer process”.

On Page 5, the third paragraph, we have replaced “indicating a process whereby the excited WSe₂ A excitons transfer into the CNT and relax to the E₁₁ state” with “indicating a process where absorption by WSe₂ A excitons results in emission from CNT E₁₁ state”

In the same paragraph, we have added the following sentence: “We hypothesize that exciton transfer process is responsible for coupling the excited and the ground states within different materials.”

Regarding the PLE polarization measurements for the CNT without the WSe₂ ML, the results are shown in Figure R1. The excitation polarization of the E₂₂ displays a significant linear polarization consistent with the literature [J. Lefebvre, *et al.* Phys. Rev. Lett. 98, 167406, 2007; D. A. Tsyboulski, *et al.* Nano Lett. 5, 975, 2005.; S. Moritsubo, *et al.* Phys. Rev. Lett. 104, 247402, 2010.] and remains almost unchanged after the formation of the heterostructure.

Figure R1: Excitation polarization dependence of the CNT emission by integrating over a 30-meV-wide spectral window centered at the E_{11} energy (red circles) before and (blue circles) after forming the (9,8) CNT/1L WSe₂ heterostructure. The excitation is adjusted to the E_{22} energies (1.55 eV for red circles and 1.51 eV for blue circles). The lines are fits to a cosine squared function.

In Line 96, the authors stated “Remarkably, a prominent high-energy peak appears at ~ 1.673 eV in the excitation dependence, which is a signature of the exciton transfer process”. This is a speculation because based on Fig. 1e, this peak corresponds to WSe₂ ML emission.

The reviewer is correct in identifying this peak as a transition corresponding to the A exciton peak in WSe₂ ML emission spectrum (gray curve in Fig. 1e), which indicates that absorption by WSe₂ results in emission from the CNT. Since both energy transfer and exciton transfer can explain this phenomenon, we should not attribute this peak to exciton transfer at this point. As mentioned above, we decided to take a more conservative standpoint and deleted the clause “which is a signature of the exciton transfer process” in this sentence.

Yes, in the presence of CNT, the peak is broader on the high energy side, which can be due to many reasons other than exciton transfer (it reveals WSe₂ excitation spectrum or absorption spectrum)(e.g. technical issues because the authors did not show the emission data at photon energies beyond 1.75 eV, dielectric screening effect in the presence of CNT, etc.)

We would like to bring to your attention the fact that we never attributed the broadening of the E_A peak on the high energy side in the PLE spectrum (green curve in Fig. 1e) to exciton transfer. On Lines 108-111, we only suggested that the broadening can be explained by the continuum of absorption in WSe₂. Regardless of the reason for the broadening, it does not change our observation that absorption in WSe₂ results in emission from CNT.

On Lines 103-105, the authors stated that “The EA peak and the E22 peak in the PLE spectra are comparable in their height, showing that the exciton transfer process is highly efficient despite the different dimensionalities”. Again, this is a speculation because the fact that the intensity of the peak remained almost the same in the presence of the CNT indicates that there is no charge transfer nor exciton transfer.

We thank the reviewer for this comment. The reviewer seems to be comparing the samples with and without CNT, *i.e.*, the E_A peak in WSe₂ PL (gray curve) and CNT/WSe₂ PLE (green curve), whereas the above sentence (Lines 106-108 in the revised manuscript) refers to the green curve only. We did not explicitly refer to the curves, which may have led to this confusion. The comparable heights of the E_A and E₂₂ peaks in the green curve indicate that the excitation through the E_A peak is as efficient as the E₂₂ peak.

To clarify which spectrum we are referring to, we have added references to the gray curve and the green curve in this paragraph. We have also added a description of the normalized intensity in the caption of Fig. 1e to avoid any confusion regarding the peak height of the gray curve being similar to the green curve.

To prove this, the authors should consider the fact that the overlap area (CNT/WSe₂) from which the emission originates is much smaller than the excitation spot size, thus, if we assume that there is exciton transfer, the emission of these transferred excitons would be much weaker than the emission of WSe₂ excitons (in the absence of CNT).

We would like to remind the reviewer that we already considered the overlap area between the CNT and the WSe₂ flake in the third paragraph of Page 9, in order to quantitatively evaluate the exciton transfer efficiency. Detailed analysis is available in Supplementary Note 8.

Briefly, the ratio of the transferred A excitons is estimated by comparing the PL emission intensity under E_A and E₂₂ excitations. The two excitation processes involve different steps, regarding their different dimensions, light absorption cross-section, and absorption coefficient. The dimensional difference between the materials is considered as follows. For 2D materials, the light within the laser spot entirely illuminates the sample. In contrast, for CNTs, given its diameter of 1.0 nm and the Gaussian laser spot radius of 0.58 μm, only 1.4×10^{-3} of light interacts with the CNT. With a simple calculation involving other steps, we have estimated that approximately 3.8% of the A excitons excited within the laser spot transfer into the CNT for the sample on resonance. For most off-resonant samples, the ratio of the transferred excitons is even lower.

To clarify this point, we have added a description of the estimation of the transferred A excitons in the third paragraph of Page 9, as “From the α factor, we estimate that approximately 3.8% of the A excitons transfer into the nanotube taking into account the different absorption areas and absorption coefficients (see Supplementary Note 8).”

In my opinion, based on Fig. 1c and 1d, it is clear that there is energy transfer from WSe₂ ML to CNT because the emission of the CNT (around 0.88 eV) increased in the presence of the WSe₂ ML.

Since both exciton transfer and energy transfer are consistent with the increased PL emission in the PLE map of Fig. 1d compared to Fig. 1c, we agree that we should take a more conservative standpoint in the earlier sections. As already noted above, we have revised the manuscript accordingly.

In my opinion, although the studied system is very interesting, the experiments are well designed, and the results are well presented, the manuscript should not be accepted in Nature Communications because the conclusions are speculative and not proved. If there are convincing results to conclude exciton transfer, I would recommend publication in Nature Comm, but the results suggest that what took place at this interface is energy transfer, which is not new taking into account that previous reports have already observed energy, and/or charge transfer at the interface of 2D-TMDs and other nanostructures such as Plasmonics, QDs, and 2D/2D.

We thank the reviewer for mentioning that “the studied system is very interesting, the experiments are well designed, and the results are well presented”. We also thank the reviewer for the insightful comment regarding a possible interpretation by energy transfer.

As explained above, we rule out the Forster-type energy transfer based on the resonant behavior as discussed on Pages 10-11. We observe large changes in the transfer efficiency factors for CNT chiralities having similar E_{22} energies as shown in Fig. 3b-e. For the energy transfer process where WSe₂ A exciton is the donor and CNT E_{22} exciton is the acceptor, the transfer efficiency should stay roughly constant as the spectral overlap does not change much. The Forster-type energy transfer is therefore inconsistent with the experimental observation. Such a behavior can be explained in terms of transfer efficiency being modulated by band alignment, where the significantly enhanced excitation at the E_A peaks for (10,5) CNT/WSe₂ heterostructures indicates the resonant exciton transfer process. The validity of the model is further examined by the density functional theory as shown in Supplementary Note 7.

We agree that we should take a more conservative standpoint in the earlier sections, and we have made changes to the main text accordingly. We now draw the conclusion on Page 11, the third paragraph, as “The theoretical findings corroborate with the empirical results described above, supporting the picture that band alignment transition is taking place and confirming the exciton transfer process.”

Response to Reviewer #3:

In this work, the authors reported the exciton transfer from two-dimensional layered WSe₂ to one-dimensional carbon nanotubes in the mixed-dimensional heterostructures. They constructed the mixed-dimensional heterostructure by transferring WSe₂ of few layers to CNTs that were grown across trenches, and measured the photoluminescence excitation (PLE) spectroscopy of the heterostructures. From time-resolved PL measurements, they found that the decay lifetime is much longer for suspended (9, 8) CNTs/WSe₂ excited by EA than that by E₂₂. The PL intensity maps also showed that the PL is spatially expanded if excited by EA than E₂₂, and a diffusion

length of excitons of 0.6 μm was estimated. The resonant exciton transfer was further studied by comparing heterostructures with CNTs of different chirality. DFT calculations of the energy band alignment were performed to explain the enhancement and suppression of exciton transfer for different CNTs. Although the exciton transfer process within CNT/WSe₂ heterostructures reported in this work is new observation, the reviewer can not recommend publication of this work in Nature Communications due to the following reasons, and possible revisions are also suggested below.

We sincerely thank the reviewer for accurately describing the results and commenting that this work is new observation.

1. The authors measured the CNT/WSe₂ mixed dimensional heterostructures of different CNTs. However, more experimental details should be provided.

As requested, we have provided more experimental details below.

2. How did the authors assign the chirality of the suspended CNTs?

We performed PLE spectroscopy to identify the chiralities of suspended CNTs by comparing E_{11} and E_{22} energies to tabulated data (A. Ishii *et al.*, Phys. Rev. B 91, 125427, 2015). Accordingly, we added the following sentence to the description on air-suspended carbon nanotube growth in the Methods section:

“PLE spectroscopy is conducted to identify the chiralities of suspended CNTs by comparing E_{11} and E_{22} energies to tabulated data [12].”

3. How did the authors ensure the contact between CNTs and WSe₂? Particularly, CNTs are across trenches, and it is natural that CNTs will bend down toward the trench. If they compare CNTs across trenches of different width, it is expectable that different results will be obtained.

We sincerely thank the reviewer for an important comment. We have confirmed the contact between CNTs and WSe₂ by examining the redshifts of E_{11} and E_{22} peaks after transferring the WSe₂ flake. As depicted in Fig. 1c,d, the heterostructure displays redshifts of 33 and 54 meV for E_{11} and E_{22} peaks, respectively, compared to the pristine CNT. Such redshifts arise from the dielectric screening effect of WSe₂. Similar redshifts induced by a 2D material were systematically explored in our previous study on CNT/h-BN heterostructures (N. Fang *et al.*, ACS Photonics 7, 1773, 2020). In that study, we analyzed various samples of h-BN and CNTs, examining their morphology with AFM and measuring the redshifts. It was observed that heterostructures with intimate contact show E_{11} and E_{22} redshifts ranging from 30-70 meV. Although the 2D material under study in this manuscript is WSe₂ and not h-BN, the comparable redshifts imply a similar intimate contact. Moreover, from the morphology data by AFM presented in response to Question #4, we can conclude that the CNTs are in contact with the WSe₂ flakes.

We also recognize that some CNTs do not make contact with WSe₂ after forming the

heterostructure. Fig R2 presents PLE maps of a (10,5) CNT before and after the transfer of a 3L WSe₂ flake. Absence of clear redshifts points to a lack of intimate contact between WSe₂ and CNT. Consequently, we have excluded these samples from subsequent study. As the reviewer kindly pointed out, we did notice that longer CNTs tend to lack contact with the WSe₂ flake after the formation of heterostructures. Hence, we've chosen relatively short CNTs with lengths ranging from 0.5 to 2.0 μm in this study. Based on the above discussion, we have added the following sentences to the Methods section:

“We note that some of the CNT/WSe₂ samples exhibit substantially small E₁₁ and E₂₂ energy shifts of less than 5 meV, suggesting a lack of contact between the CNT and the WSe₂. These samples are excluded in the following study.”

Fig. R2 PLE maps of a (10,5) CNT before (left) and after (right) the transfer of a 3L WSe₂ flake which did not show much redshifts.

4. The authors should provide morphology data, such as SEM or AFM, where CNTs are visible.

As requested, we have conducted measurements of our CNT/WSe₂ heterostructures using both AFM and SEM. For SEM, a single suspended CNT is difficult to identify and only bundles can be resolved. As shown in Fig. R3, a Y-junction CNT bundle was observed in a pristine sample. For the heterostructure sample, the CNTs beneath the WSe₂ flake were not observed although we carefully scanned the sample under SEM.

Using AFM, we have successfully identified CNTs beneath the WSe₂ flake, as illustrated in Fig. R4. Multiple CNT tubes and Fe catalyst can be seen above the substrate, which contributes to the rough surface of WSe₂ on the substrate. On the other hand, the suspended part of the WSe₂ flake exhibits uniformity characterized by minimal surface roughness, highlighting the cleanliness of our transfer process. Furthermore, this suspended section falls into the trench by approximately 6 nm, resulting in an intimate contact with the CNTs. As a result, two CNTs are

discernible across the trench, as marked by arrows in Fig. R4b. The height of the upper CNT is roughly 1.05 nm, which is consistent with the CNT diameter discussed in this paper. We therefore have added the AFM images as Supplementary Fig. 1 and have added one sentence about the morphology in the first paragraph of Page 5:

“The morphology of the heterostructure is examined with an atomic force microscope (Supplementary Fig. 1), and an intimate contact between the CNT and the WSe₂ is confirmed.”

Fig. R3 SEM images of a suspended CNTs (left) and a CNT/3L WSe₂ heterostructure (middle, right). The white circle in the middle image indicates the measured area for the right image. The scale bars are 2, 2, 1.5 μm, respectively, from left to right.

Fig. R4 (left) An optical microscope image of a CNT/2L WSe₂ heterostructure. 2D (middle) and 3D (right) AFM images for the sample. The scale bars are 5 and 0.4 μm for the left and middle images, respectively.

5. Raman spectroscopy could be possible to confirm the interaction between CNTs and WSe₂?

We thank the reviewer for the suggestion of using Raman spectroscopy. Although Raman measurements can provide insights on coupling of phonons in the heterostructure, there are no established results available to confirm the interactions between CNTs and WSe₂. Since the primary focus of this manuscript is on excitonic physics, we feel that investigating the influence

of heterostructure formation on phonon spectra is beyond the scope of the current manuscript. We agree that it would indeed be an interesting research topic for the next step.

6. Did the authors compare the PLE of the heterostructure for suspended CNTs and supported CNTs on the substrate and what is the difference?

As shown in Fig. 2e,f, CNTs directly in contact with the substrate do not show emission, and therefore it is not possible to perform PLE measurements in heterostructures where WSe₂ is on top of the CNTs.

We therefore prepared new heterostructures, where WSe₂ is below the CNTs for supporting the tubes. The optical microscope image is shown in Fig. R5 (left, top), with WSe₂ flakes first being prepared on a SiO₂/Si substrate. CNTs are then transferred onto the WSe₂ flakes by using the anthracene-assisted transfer technique (K. Otsuka *et al.*, Nat. Commun., 12, 3138, 2021). The PLE of the heterostructure for supported CNTs on the WSe₂/SiO₂ substrate are consistent with the suspended CNT/WSe₂ in the current manuscript, showing similar exciton transfer behavior as described below.

Fig. R5 (left, top) An optical microscope image and (left, bottom) a reflectivity image of the CNT/WSe₂ heterostructure. (right) PLE maps from five positions, which are marked as white circles in the reflectivity image. The excitation power is 10 μ W. The scale bars are 20 μ m.

By scanning the sample with PL microscopy, we found 5 bright tubes on top of the WSe₂ flake. Their positions are marked in the reflectivity image (left, bottom), which are labelled as P1, P2, and so on. As shown in Fig. R5 (right), PLE maps from these positions show different E₁₁ emission energies, indicating different chiralities of the CNTs. High-energy excitation peaks are observed around 1.65 eV for the PLE from P1, 3, 4, and 5. Such excitation energy dependence is consistent with the PLE map in Fig. 1d and PLE spectra in Fig. 3, which is consistent with exciton transfer process. We note that the PL intensity is much weaker than that from the suspended heterostructures, possibly owing to the shorter length of the transferred CNTs. Among them, the PLE map from P3 exhibits the highest intensity. We assign the chirality to (10,5) since both the emission energy and the excitation energy dependence are similar to that of the suspended (10,5) CNT/WSe₂ heterostructure in the main text. (10,5) CNTs have been demonstrated to exhibit a resonant exciton transfer process in Fig. 3, which could account for the strong PL emission from P3.

7. For the PL maps shown in Figure 2 and 4, what is the wavelength range for the signal was collected? Is it PL from WSe₂ or CNTs, or both?

We thank the reviewer for pointing out the insufficient experimental description. We used a longpass filter with a cut-on wavelength of 1050 nm to only collect PL emission from CNTs. The PL images are constructed by integrating PL emission over a 30-meV-wide spectral window centered at the E₁₁ energy (Fig. 2) or over the full spectrum (Fig. 4). Accordingly, we have revised the caption for Fig. 2, 4 and the following two sentences in the paragraph for Photoluminescence measurements in the Methods section:

“A custom-built confocal microscopy apparatus is utilized to perform PL measurements for CNT E₁₁ excitons at ambient temperature within an atmosphere of dry nitrogen gas [12, 23].”

“A longpass filter with a cut-on wavelength of 1050 nm is used to collect the PL emission from CNTs.”

8. What the excitation wavelength for the PL of WSe₂ shown in Figure 1e? Do they expect PL intensity of WSe₂ using EA excitation?

As noted in Methods (Line 354), we employ a 532-nm laser for the measurement of WSe₂ PL spectrum in Fig. 1e. We do expect PL emission from WSe₂ at E_A excitation, however, as mentioned in the response to Question 7, the use of a longpass filter prevents the collection of PL emission from WSe₂. In principle, such resonant PL measurement is possible, but it is experimentally challenging since the excitation laser scatter can overwhelm the emission signal, requiring special techniques for example crossed polarization configuration or non-collinear configuration to avoid pump scatter.

9. The authors used a collection spot size of 5.4 um in diameter as they described in the experimental section, how could the CNTs in Figure 4a and 4b be resolved spatially?

We thank the reviewer for the comment. As addressed in the response to Question 2 for Reviewer

#1, CNTs can be resolved because the laser spot diameter of 1.16 μm is much smaller than the collection spot diameter of 5.4 μm and the image resolution here is determined by the laser spot size. Emission outside the excitation laser spot is also detected because of the large collection spot, and therefore the images represent excitation efficiency profiles for CNT PL.

To clarify this point, we have added a description of the PL excitation imaging measurement and laser spot size in the second paragraph of Page 8 and in the Methods section of photoluminescence measurements, as noted in our response to Reviewer #1.

10. Two (10, 5) CNTs are shown in Figure 4a, but the intensities are apparently different, can the authors explain the difference?

The difference in PL intensity between the two (10,5) CNTs is primarily due to their different lengths. As depicted in the revised version of Fig. 4a, b, the brighter (10,5) tube, with the different orientation, is longer in length. Therefore, this tube experiences a reduced end quenching effect, and the PL emission is stronger. The length dependence of the CNT PL intensity can be found in our previous work (A. Ishii *et al.* Phys. Rev. B 91, 125427, 2015). We therefore have added a sentence in the first paragraph of Page 13:

“We note that the two (10,5) CNTs exhibit different PL intensities, which is primarily due to their different lengths [12].”

Reviewers' Comments:

Reviewer #1:

Remarks to the Author:

The authors have addressed the concerns and comments by reviewer, I would recommend for publication of present revision.

Reviewer #2:

Remarks to the Author:

Dear Editor,

The authors have addressed my concerns, in particular, in the Revised Manuscript, they have used a conservative approach to make conclusions. They addressed the possibility of energy transfer in addition to exciton transfer to interpret the observations.

In my opinion, they Revised Manuscript can be published in Nature Comm.

Sincerely,

Reviewer #3:

Remarks to the Author:

In the revised manuscript, the authors added sufficient experimental details for their measurements, and additional data of the morphology characterization. The revised manuscript is now more clear and can be recommended for publication in Nature Communications after three minor revisions.

1. In Fig 1f, the authors should mark the orientation of the trench (0 degree) either in the figure or in the caption.
2. In Fig 3f, the authors should mark the difference of the black arrow in the schematic of Type-I and red arrow in Resonance, although they discussed in the maintext.
3. Please also mark the positions of CNTs in the PL images in SI.

Response to reviewer reports for manuscript No. NCOMMS-23-39012A by Fang *et al.*

Reviewer #1 (Remarks to the Author):

The authors have addressed the concerns and comments by reviewer, I would recommend for publicaiton of present revision.

We are grateful for the valuable feedback that helped us improve our manuscript. We are very happy to hear that the reviewer suggests the publication of our work.

Reviewer #2 (Remarks to the Author):

Dear Editor,

The authors have addressed my concerns, in particular, in the Revised Manuscript, they have used a conservative approach to make conclusions. They addressed the possibility of energy transfer in addition to exciton transfer to interpret the observations.

In my opinion, they Revised Manuscript can be published in Nature Comm.

Sincerely,

We thank the reviewer for the careful evaluation of our revised manuscript, and for recommending our manuscript for publication in Nature Communications.

Reviewer #3 (Remarks to the Author):

In the revised manuscript, the authors added sufficient experimental details for their measurements, and additional data of the morphology characterization. The revised manuscript is now more clear and can be recommended for publication in Nature Communications after three minor revisions.

We appreciate the helpful comments and the recommendation for publication in Nature Communications after minor revisions. Below, we have revised our manuscript point-by-point following the reviewer's suggestions.

1. In Fig 1f, the authors should mark the orientation of the trench (0 degree) either in the figure or in the caption.

As suggested, we have added one clause to describe the orientation of the trench in the caption of Fig. 1: “where 0 degrees correspond to the direction along the trench.”

2. In Fig 3f, the authors should mark the difference of the black arrow in the schematic of Type-I and red arrow in Resonance, although they discussed in the maintext.

We thank the reviewer for bringing up this point. We have added the explanation of the black and the red arrows in the caption of Fig. 3:

“Black arrows indicate the exciton tunneling process, while a red arrow indicates the resonant tunneling of holes when the valence bands match.”

3. Please also mark the positions of CNTs in the PL images in SI.

As requested, we have revised Supplementary Figure 5 to mark the CNT position and orientation.